# Caffeic Acid Phenethyl Ester Protects Neurons Against Oxidative Stress and Neurodegeneration During Traumatic Brain Injury

**DOI:** 10.3390/biom15010080

**Published:** 2025-01-08

**Authors:** Nurul Sulimai, Jason Brown, David Lominadze

**Affiliations:** 1Department of Surgery, University of South Florida Morsani College of Medicine, Tampa, FL 33612, USA; nurulsulimai@usf.edu (N.S.); jasonb3@usf.edu (J.B.); 2Department of Molecular Pharmacology and Physiology, University of South Florida Morsani College of Medicine, Tampa, FL 33612, USA

**Keywords:** mild-to-moderate traumatic brain injury, neurons, nuclear factor kappa B, reactive oxygen species

## Abstract

Traumatic brain injury (TBI) is an inflammatory disease causing neurodegeneration. One of the consequences of inflammation is an elevated blood level of fibrinogen (Fg). Earlier we found that extravasated Fg induced an increased expression of neuronal nuclear factor kappa B (NF-κB) p65. In the present study, we aimed to evaluate the effect of caffeic acid phenethyl ester (CAPE), an inhibitor of NF-κB, on Fg-induced neurodegeneration in vitro and in mice with mild-to-moderate TBI. Primary mouse brain cortical neurons were treated with Fg (0.5 or 1 mg/mL) in the presence or absence of CAPE. A cortical contusion injury -induced model of TBI in C57BL/6 mice was used. Mice were treated with CAPE for two weeks. The generation of reactive oxygen species (ROS) and neuronal viability were assessed. Mice memory was assessed using novel object recognition and contextual fear conditioning tests. The generation of ROS and viability of neurons in vitro and in the brain samples were assessed. Data showed that CAPE attenuated the Fg-induced generation of ROS and neuronal death. CAPE improved the cognitive function of the mice with TBI. The results suggest that Fg-induced generation of ROS could be a mechanism involved in cognitive impairment and that CAPE can offer protection against oxidative damage and neurodegeneration.

## 1. Introduction

Traumatic brain injury (TBI) is an inflammatory neurodegenerative disease that affects the lives of people of all ages. Inflammation is accompanied by an increase in the level of acute-phase reactant proteins such as fibrinogen (Fg) [1,2]. We found that, at higher levels, Fg interacts with endothelium increasing vascular permeability and resulting in Fg transmigration to the interstitium in mice with mild-to-moderate TBI (m-mTBI) [3]. Furthermore, extravascular deposits of Fg have been found in post-mortem brain samples of patients with TBI [4]. When Fg interacts with its astrocytic and neuronal receptors, intercellular adhesion molecule-1 (ICAM-1) and cellular prion protein (PrP^C^), Fg itself can induce an inflammatory reaction [5,6]. We have demonstrated that Fg caused an increase in the level of pro-inflammatory cytokines C-X-C motif chemokine ligand 10 (CXCL10) and C-C motif chemokine ligand 2 (CCL-2) in astrocytes and upregulated interleukin-6 (IL-6) in astrocytes and neurons [5,6]. In addition, at an elevated level, Fg increased the expression of neuronal nuclear factor-κB (NF-κB) p65, oxidative damage, and the death of astrocytes and neurons [5,6].

NF-κB is a ubiquitous transcription factor and plays a critical role in regulating gene expressions related to stress and inflammation in the nervous system [7,8,9,10]. The activation of NF-κB is linked to various neurodegenerative disorders, including TBI [7,11,12]. Prolonged activation of NF-κB during [12,13] TBI has been shown, but the specific cause of its post-traumatic activation was not clarified [12]. We have shown that the use of caffeic acid phenethyl ester (CAPE), a specific inhibitor of NF-κB p65, ameliorated the Fg-induced increase in NF-κB p65 expression and the upregulation of *CCL-2* and *IL-6* in the cultured neurons [14]. These results suggested the possibility that m-mTBI-induced neurodegeneration may involve the interaction of Fg with neurons via the activation of the transcription factor NF-κB. CAPE has been reported to have anti-inflammatory, antioxidant, neuroprotective, and immunomodulatory properties in various systems, with a potential to improve neurodegeneration during m-mTBI [15,16,17]. We found that there was an increased expression of NF-κB p65 in the neurons of mice with cortical contusion injury (CCI) compared to that in sham-operated mice [13]. However, the transgenic mice with a lower blood level of Fg demonstrated less increase in the expression of neuronal NF-κB p65 after CCI than wild-type (WT) mice [13]. Others have shown that CAPE protects blood–brain barrier integrity (BBB) and reduces contusion volume in rat models of severe TBI [18]. However, despite the effect of CAPE in the mitigation of Fg-induced neurodegeneration, its benefits during m-mTBI, in vitro and in vivo, have not been shown. In the present study, we aim to explore the neuroprotective potential of CAPE in Fg-induced oxidative damage of neurons during m-mTBI.

## 2. Materials and Methods

### 2.1. Antibodies, Reagents, and Materials

The following reagents and materials were purchased from Thermo Fisher Scientific (TFS) (Waltham, MA, USA), including primary mouse brain cortex neurons (PMBCNs) from C57BL/6 mice (cat. #A15586), Nunc™ 12-well plates (cat. #150628), Nunc™ 24-well plates (cat. #142475), German #1 glass coverslips (cat. #50-121-5159), phosphate-buffered saline (PBS) (composition: 1.05 mM of KH_2_PO_4_, 155.17 mM of NaCl, and 2.97 mM of Na_2_HPO_4_. 7H_2_O, without calcium (Ca^2+^) and magnesium (Mg^2+^)), Hank’s Balanced Salt Solution (HBSS), containing Ca^2+^ and Mg^2+^ (cat. # 14025076), dimethylsulfoxide (DMSO) (Fisher, cat. #50-188-278FP), ACROS Organics™ Triton™ X-100 (cat. #AC422355000), normal goat serum (cat. #NC9660079), and Alexa Fluor-conjugated secondary antibodies (cat. #A11032). The growth media and the additive for the PMBCN culture were purchased from Lonza (Basel, Switzerland). This was used as the neuron basal medium (cat. #CC-3256) with a specific additive, SingleQuots^TM^ (cat. #CC-4462). The poly-D-lysine (PDL) hydrobromide (cat. #P0899) and laminin from Engelbreth–Holm–Swarm murine sarcoma basement membrane (cat. #L2020-1MG) were purchased from Millipore-Sigma (Burlington, MA, USA). Other materials purchased from Millipore-Sigma were hirudin from leeches (cat. #H7016-10UN), bovine serum albumin (BSA) (cat. #A7906-10G), and primary antibodies against NF-κB p65 (cat. #06-418) and neuronal nuclei (NeuN) (cat. #MAB377). The inhibitor of NF-κB p65 activity, CAPE (for in vitro study), was purchased from R&D systems (Minneapolis, MN, USA, cat. #2743/10). CAPE for in vivo administration was purchased from Selleck Chem (Houston, TX, USA, cat. #S7414).

The polyclonal rabbit antibody against human Fg was purchased from Dako Cytomation (Carpentaria, CA, USA, cat. #A008002-2). ProLong™ Diamond antifade mounting media with 4′,6-diamidino-2-phenylindole was purchased from Invitrogen (Carlsbad, CA, USA, cat. # P36962).

### 2.2. Cultured Neurons

The PMBCNs were plated at 7 × 10^5^ cells/mL and grown on 24-well plates with German #1 glass coverslips or oncell culture plates that were first coated with PDL hydrobromide (30 μg/mL) and laminin (200 μg/mL) for 1 h at room temperature (RT), and then rinsed twice with PBS without Ca^2+^ and Mg^2+^ before cell seeding. The PMBCNs were cultured in neuronal basal medium with the additive SingleQuots^TM^ and were used on day 7 or 10.

The neurons were grown and treated as we have reported previously [6,14]. Prior to treatment with Fg, the neurons were pre-treated with CAPE at 25 µg/mL for 90 min. Fg was added to the neurons for 1 h. In the control group, an equal volume of PBS was used instead of Fg. Each experimental group contained hirudin (0.5 U/mL) to block the possibility of thrombin-induced conversion of Fg into fibrin. The cells were kept in an incubator at 37 °C 5% CO_2_.

### 2.3. Detection of Reactive Oxygen Species (ROS)

Generation of neuronal intracellular ROS was detected using a carboxy-H2DCFDA Image-IT™ LIVE Green ROS Detection Kit (Invitrogen, Carlsbad, CA, USA), following the manufacturers’ recommendations and our previously published protocol [6]. Carboxy-H2DCFDA is cell-permeable and becomes fluorescent when intracellular esterases remove the acetate groups from the compound and oxidation occurs within the cells. The PMBCNs were incubated with carboxy-H2DCFDA at a final concentration of 30 μM in HBSS, containing Ca^2+^ and Mg^2+^, at 37 °C, for 30 min in the dark, followed by three 5 min washes with PBS. Fluorescence intensity was measured at the wavelength of 494/529 nm (λexcitation/λemission). The mean fluorescence intensity per cell was presented after subtraction of the background fluorescence.

The detection of ROS ex vivo was performed using dihydroethidium (DHE), which is cell-permeable and is oxidized to ethidium by the action of superoxide, which produces red fluorescence. DHE (Invitrogen, cat. #D11347) was dissolved in DMSO to make 10 mM stock and further diluted in PBS to a final concentration of 2 µM. Brain slices collected, flash-frozen, and sectioned to 25 µm thickness on a microscope slide were incubated with DHE in a humidified chamber protected from light for 30 min at 37 °C. Sections were then rinsed in PBS and fixed in paraformaldehyde (PFA) for 20 min at RT before proceeding with immunohistochemistry.

### 2.4. Cell Live/Dead Assay

The PMBCNs were treated as described above with 0.5 mg/mL of Fg (Fg 0.5) or 1 mg/mL of Fg (Fg1), in the presence or absence of CAPE or PBS. PMBCNs treated with PBS or CAPE alone served as controls. After treatment, the cells were rinsed with HBSS (with Ca^2+^ and Mg^2+^) at 37 °C before proceeding with the LIVE/DEAD^®^ Viability/Cytotoxicity Assay (Fisher, cat. #L3224). The live and dead cells were identified by simultaneously activating green-fluorescing calcein-AM (494/517 nm), which indicated the presence of intracellular esterase activity, and red-fluorescing ethidium homodimer-1 (EthD-1) (528/617 nm), which signified dead cells identifying compromised plasma membrane. Calcein-AM and EthD-1 were used at working concentrations of 2 and 2.5 µM, respectively, diluted in HBSS (with Ca^2+^ and Mg^2+^) and stained as recommended by the manufacturer.

### 2.5. Cortical Contusion Injury

The procedures performed and the use of animals for the study adhered to the National Institutes of Health’s Guide for the Care and Use of Laboratory Animals and has been approved by the University of South Florida’s Institutional Animal Care and Use Committee (R10673). Ten- to twelve-week-old (26–30 g weight) WT, C57BL/6J male and female mice, weighing 26–30 g, were used. Mice were anesthetized with isoflurane and placed in a stereotaxic frame on a heating pad. A skin incision was performed mid-section on the cranium before performing a 4 mm diameter craniotomy, using a microrongeur (Fine Science Tools, Foster City, CA, USA) on the left hemisphere at −2.5 mm bregma and 2.75 mm lateral to the midline. We induced an m-mTBI using a well-established CCI model, as we previously described [3,13,19]. In short, the CCI was produced with a 2 mm diameter flat-tip impactor at the speed of 3.5 m/s, with a depth of 0.5 mm. Following the impact, the skull cap was replaced with Surgicel (Johnson & Johnson, New Brunswick, NJ, USA, cat. #136215) before the skin was sutured. The sham-operated group underwent the same procedure excluding the actual impact with the impactor.

### 2.6. CAPE Administration

CAPE was prepared in 1% of DMSO and further diluted 1:100 in Lactated Ringer solution (LRS). The dose of administered CAPE was 10 mg/kg. The CAPE was administered at 10 mg/kg (given intraperitoneally, IP). As a control, a vehicle solution (VEH, 1% DMSO in LRS) did not contain CAPE. The first doses of CAPE or VEH were given 30 min after the CCI and then continued daily for 14 days.

### 2.7. Blood and Brain Sample Collection

Fourteen days after CCI or sham operation, blood and brain samples were collected. Mice were anesthetized with Ketamine (87.5 mg/kg)/Xylazine (12.5 mg/kg) injected intraperitoneally. A midline abdominal incision was performed to reach the caudal vena cava. An estimated 600–800 µL of blood was withdrawn using a 1 mL syringe and a 22G needle pre-filled with 50–60 µL of anticoagulant buffer, 3.2% sodium citrate in PBS. After exsanguination, cervical decapitation was performed, and the brain was collected immediately by the flash-freezing method [20]. The brain was quickly removed from euthanized mice and dipped for 1 min in isopentane pre-chilled with liquid nitrogen. The frozen brain was then transferred to a plastic embedding mold. The cryomold was filled with optimal cutting temperature compound (OCT) (Fisher, cat. #23-730-571) and, using forceps, placed floating in the isopentane bath in the liquid nitrogen until the OCT solidified. The sample was kept at −70 °C until it was ready for cryosectioning.

### 2.8. Immunohistochemitry

For each experimental group, brain slices from at least 6 separate mice were analyzed. The fixed brain samples were stained for NeuN (Millipore-Sigma), as recommended by the manufacturer. Epitope retrieval with 0.01 M Citrate Buffer, pH 6.0, was performed before immunostaining [21]. The slides were immersed in the buffer, boiled to 100 °C for 20 min, and then rinsed with distilled water. Then, the brain samples were blocked with a blocking buffer (0.3% TritonX-100, 5% donkey serum in PBS) for 1 h at RT. The slides were incubated with a primary antibody, mouse anti-NeuN (1:100), at 4 °C overnight. After washing with PBS three times, for 5 min each, the tissues were incubated with Alexa Fluor-conjugated secondary antibody, donkey anti-mouse 488 (Invitrogen, cat. #A21202) (1:200), for 2 h at RT.

### 2.9. Image Analysis

For the in vitro ROS and Live/Dead assay, images were taken using an Olympus IX51 fluorescence microscope (Olympus Corporation, Tokyo, Japan). The microscope settings were kept constant for each experiment for an adequate comparison. For the ROS assay, the fluorescence was read at 485 nm excitation (Ex) and 528 nm emission (Em) wavelengths in each microscopic field. Fluorescence intensity in eight randomly placed fixed-size areas of interest was measured. The results were expressed as the average of the fold increase in fluorescence intensity over the untreated control. The measurement of fluorescence was performed using Olympus CellSens Dimension Desktop 2.3 (Olympus Corporation). For the Live/Dead assay, live cells appear green while dead cells appear red with the wavelengths Ex/Em = 489–495/530–635 nm for each, respectively. Cell counting was performed using CellSens Dimension Desktop 2.3. The number of live and dead cells was counted as the total number. The results were presented as a percentage of the number of live cells of the total cell count.

For ex vivo ROS detection, the images were taken with an Olympus FV1200 (Tokyo, Japan) confocal microscope. Image analyses were performed using CellSens Dimension Desktop 2.3. The DHE and NeuN fluorescence wavelengths were Ex/Em = 520–610/498–522. The results were presented as the count of NeuN-positive cells and the count of DHE-positive cells co-localizing with NeuN.

### 2.10. Novel Object Recognition Test (NORT) and Contextual Fear Conditioning Test (CFCT)

The NORT is a commonly used behavioral test for investigating various aspects of learning and memory in mice. The details of the experimental procedure were previously described [19,22,23]. One day before the test day, mice were placed individually in an empty testing chamber for a 10 min habituation session and returned to their home cage. On the day of the experiment, mice were placed in the testing chamber for 10 min with two identical objects (acquisition session) and returned to their home cages for 1 h. Mice were then returned to the testing chamber in the presence of one of the original objects and one novel object (recognition session) for 5 min. The movements of the mice were recorded using a webcam that was positioned to observe from an aerial view of the testing chambers. Between the tests, the chambers and objects were sanitized using 70% ethanol (diluted in water). Exploratory behavior was characterized by actions such as sniffing, touching, and focusing attention on the object. A “memory score” for each mouse was presented as the discrimination index (DI), which was determined using the following formula DI  =  (Novel Object Exploration Time/Total Exploration Time)  −  (Familiar Object Exploration Time/Total Exploration Time) [24]. The data for the behavioral assessment were expressed by calculating the DI. A lower DI indicates less time spent with the novel object, thus indicating short-term memory impairment [24,25].

CFCT was performed to assess hippocampal function and memory formation. The protocols performed were described elsewhere [26,27]. On day 1, mice were trained and conditioned to the contextual cues and the auditory cues that were paired with electrical shocks delivered to their feet. The 25 × 25 cm sound attenuation chamber with a wire grid floor was cleaned with 70% ethanol prior to and between the procedures. We verified that the shock was delivered to the floor grid by touching it with our fingers, the auditory cue was generated, and the camera was positioned and focused before the testing. Mice were allowed to explore the context (the chamber and its environment) for 3 min, then they received the conditioned stimulus (CS), where they heard a 90 db tone for 30 s. At the end of the 30 s, mice received a mild foot shock (0.5 mA) that lasted for 2 s, which serves as the unconditioned stimulus (US). After 3 min, the mice received a second CS/US pairing. Twenty-four hours after training, mice were placed back into the chamber and allowed to explore for 3 min without tone or shock. Freezing was assessed using a weight transducer system (Panlab, Barcelona, Spain). Mice were defined as freezing if their movement ceased for at least two consecutive seconds. Data were presented as a percentage of the freezing time (PFT) of the overall test duration. For both NORT and CFCT mice, behavior video tracking, recording, and analyses were performed using ANY-maze Version 7.44 (Stoelting Co., Wood Dale, IL, USA).

### 2.11. Statistical Analysis

The data were analyzed using Graph Pad Prism 10 (Version 10.4.0 (621)) software (San Diego, CA, USA). All values are presented as mean ± SE. The experimental groups were compared using a one-way ANOVA followed by a pairwise comparison using Tukey’s multiple comparisons test. Differences between the data were considered significant at a *p* value less than 0.05.

## 3. Results

Fg dose-dependently increased the generation of ROS in cultured neurons (Figure 1A,B). Pre-treating the neurons with CAPE significantly reduced the effect. In addition, Fg dose-dependently increased neuronal death (Figure 2A,B). This Fg-induced cell death was reduced when neurons were pre-treated with CAPE (Figure 2A,B).

We observed a higher deposition of Fg in the cortical brain samples of mice with CCI compared to that in samples from sham-operated mice for both CAPE- and VEH-treated groups (Figure 3A,B). However, CAPE did not have an effect on the deposition of Fg in brains from mice with CCI or sham operation (Figure 3A,B). The higher number of spots with co-localization of Fg and DHE-positive cells were found in mice with CCI compared to those in sham-operated mice treated with VEH (Figure 3A,C). However, the number of co-localized Fg and DHE spots in CAPE-treated animals with CCI was not different from that in samples from the sham-operated mice treated with CAPE (Figure 3A,C). In the samples from mice that received CCI, there was significantly higher Fg/DHE co-localization in the VEH-treated group compared to that of samples from CAPE-treated mice (Figure 3A,C).

Significantly fewer NeuN-positive cells were found in the cortical brain sections of mice treated with VEH 14 days after CCI compared to those in sham-operated mice (Appendix A). However, we found no difference in the count of NeuN-positive cells in the cortical brain samples from CAPE-treated mice after CCI compared to that in the mice with sham injury (Appendix A).

There was a higher co-localization of DHE/NeuN in the samples from mice with CCI compared to that in the samples from sham-operated mice that were treated with CAPE or VEH (Appendix A). However, CAPE treatment significantly reduced the co-localization of the DHE/NeuN in the sham-operated mice (Appendix A).

NORT and CFCT showed memory deficits in mice with CCI (Figure 4A and Figure 4B, respectively). The NORT indicated a significant reduction in the short-term memory of the CCI mice that received VEH compared to that in sham-operated mice treated with VEH (Figure 4A). After treating mice with CAPE, we found no significant difference between the short-term memory of mice with CCI and with that in mice with a sham operation (Figure 4A). However, short-term memory was improved in mice with CCI after a treatment with CAPE compared to that in mice with a CCI treated with VEH. CFCT indicated that there was a significantly greater memory deficit in mice (both with sham operation and CCI) that received VEH compared to the respective groups of mice that received CAPE (Figure 4B). The mice with CCI that received VEH demonstrated less PFT than mice with a sham operation (Figure 4B). Treatment with CAPE still resulted in lower PFT in mice with CCI compared to that in mice with sham operation (Figure 4B).

## 4. Discussion

TBI leads to profound disability as it causes impairments in attention, learning, memory, and higher-order executive functions [28,29,30,31]. Despite extensive efforts, there are limited therapeutic options that exist to prevent or reverse cognitive dysfunction during m-mTBI. Understanding the molecular mechanism behind impaired cognition is crucial for tackling this challenge. The injuries attained during m-mTBI produce inflammation that may extend beyond the acute phase. It has been shown that m-mTBI causes prolonged activation of NF-κB [12] and memory impairment [28,29,31]. It was found that NF-κB DNA binding activity in the injured cerebral cortex increased, with its peak binding activity three days after injury, and subsided in 10 to 14 days after the injury [11]. Immediately after injury (from 4 to 24 h after the impact), NF-κB activity was mainly observed in neurons of the affected cortex as well as in astrocytes located in the corpus callosum adjacent to the injury [7,12]. It has been shown that upregulation of the NF-κB activity during m-mTBI occurs simultaneously with the upregulation of pro-inflammatory cytokine IL-6 and ICAM-1 [32]. However, the authors did not identify the cause of the pro-inflammatory reaction [32]. We have demonstrated that Fg caused the upregulation of pro-inflammatory cytokines in astrocytes [5] and in neurons [6]. These pro-inflammatory effects were partly due to the association of Fg with its receptors ICAM-1 and PrP^C^ on the surfaces of astrocytes and neurons, ultimately resulting in an increased death of these cells. We were the first to show that Fg caused an upregulation of neuronal expression of NF-κB p65 [14]. In the same study, we found an increased co-localization of neurons and NF-κB p65 in the cortex of mice. In addition, we showed that the mice with a low blood content of Fg demonstrated reduced NF-κB p65 expression and cerebrovascular permeability 14 days after CCI [13]. These findings emphasize the role of Fg and its interaction with its neuronal receptors in the activation of NF-κB p65, a key transcription factor involved in immune and inflammatory responses [14]. Our findings suggested that targeting Fg and its interaction with neurons could potentially modulate NF-κB p65 activity. In the present study, we investigated the specific role of NF-κB p65 in Fg-induced neurodegeneration in vitro (cultured neurons) and in vivo (mouse model of m-mTBI).

In addition to inflammation, oxidative stress plays a crucial role in the pathogenesis of m-mTBI by contributing to secondary injury mechanisms [33]. Oxidative stress is the biochemical and physiological stress caused by free radicals that can attack cell components [34]. We found that an inflammation-induced elevated blood level of Fg resulted in the extravascular deposition of Fg, causing an increase in the generation of ROS and nitric oxide (NO) in astrocytes [5] and increased ROS, NO, and mitochondrial superoxide in neurons [6]. The long-lasting NF-κB activation that occurs during TBI might contribute to the prolonged inflammation and neurodegeneration seen in the delayed pathology of m-mTBI. To validate this hypothesis, it was imperative to test the effects of neutralizing NF-κB activation on Fg-induced oxidative damage and neurodegeneration during m-mTBI. To block the activation of NF-κB p65, we used CAPE, which is a potent inhibitor of NF-κB and has demonstrated anti-inflammatory effects [15,16]. We have previously shown that CAPE ameliorated Fg-induced the upregulation of NF-κB p65 activity and reduced the overexpression of pro-inflammatory cytokines *IL-6* and *C-C chemokine ligand-2* in the cultured neurons [14]. Others have shown that CAPE was able to reduce oxidative stress, BBB permeability, and contusion volume during TBI [17,18,35]. However, the direct effect of CAPE on neuronal viability in mouse models of TBI remained unexplored.

In our study, we observed that Fg caused a dose-dependent increase in the production of ROS and promoted cell death in cultured neurons, as shown in Figure 1 and Figure 2. However, when the neurons were pre-treated with CAPE, the Fg-induced increase in ROS production and neuronal death were mitigated. These effects suggested that CAPE had a potent antioxidant effect that protected against Fg-induced oxidative damage in neurons.

It has been shown that CAPE was able to reduce infarct volume and apoptosis in the brain after cerebral ischemia [16]. However, the study did not clearly identify or differentiate the apoptotic cells based on their specific type within the brain. Our present study identifies neurons as the cells that are affected by CAPE to prevent the generation of ROS and cell death normally induced by an elevated level of Fg. We further investigated whether the neuroprotective effect observed in vitro would also apply in vivo in the mouse model of m-mTBI. The difference found in the number of NeuN-positive cells observed in the cortical brain sections of mice 14 days after CC and those in sham-operated mice was ameliorated by CAPE (Appendix A). NeuN is a neuron-specific nuclear protein in vertebrates and the loss of NeuN is suggestive of neuronal death [36]. Here, we show for the first time that CAPE was able to ameliorate the neuronal death during m-mTBI (Appendix A). While others have demonstrated its antioxidant role during TBI, the direct effect of CAPE on neurons itself had not been investigated previously [16,17,18]. Here, we assert that the protective effect of CAPE during m-mTBI was directly linked to neurons and their viability.

Our findings revealed that mice with CCI who received CAPE exhibited significantly better cognitive performance in comparison to the mice with CCI that received VEH (Figure 4). In contrast to our findings, Zhao et al. found that although CAPE administration reduced contusion volume and protected BBB integrity in the rat model of TBI, it did not improve performance in memory function as tested using the Morris water maze (MWM) and an associative fear memory task [18]. In our study, we used NORT and CFCT, which are known tests for learning and memory, and we evaluated short-term memory. The MWM test used by Zhao et al. is a test for spatial and long-term memory [37,38]. The difference in the cognitive assays used in our study and by Zhao et al. [18] the reason for the discrepancy between our findings. Another possible reason for the discrepancy between our findings and the findings of Zhao et al. may be due to the difference in the severity of the TBI in the models used and in the duration of the CAPE administration. Zhao et al. generated severe TBI in rats and administered the CAPE at 10 mg/kg for a total of 5 days [18], whereas we administered a similar dose of CAPE for 14 days to mice with m-mTBI. It has been shown that the protective effect of CAPE is dose- and time-dependent [16]. The CAPE dose that was administered, between 1 and 10 mg/kg, inversely correlate with the infarct volume and neurological score in a rat model of cerebral ischemia [16]. The longer duration of CAPE administration in our study probably has a better protective effect due to the longer lasting effect of the drug. In our tests, cognitive performance was improved in mice with CCI treated with CAPE compared to the mice treated with VEH. These results suggest that CAPE positively has an effect on the short-term memory of animals. A more pronounced effect of CAPE in cognitive improvement was seen in CFCT compared to that in the NORT (Figure 4A,B). In CFCT, the level of cognitive performance was better in the CAPE-treated mice, both with CCI and sham operation, compared to that in the VEH-treated mice with CCI or sham operation. CFCT also demonstrated that the sham-operated mice that received CAPE performed better than the sham-operated mice that received VEH. These results may indicate the effect of CAPE on improving the viability of neurons in the areas of the brain that are involved in the fear conditioning test assessment. Although both the hippocampus and cortical areas of the brain are essential for recognizing objects and contextual processing, the CFCT is primarily used to investigate fear-related behaviors, whereas NORT is not paired with fear [39]. In sham-operated animals, there was minimal damage to the cortex due to the absence of a traumatic impact to the brain. However, mice were still handled and went through the cranial window preparation surgery, resulting in some insignificant inflammatory responses. While NORT did not detect the effects of CAPE on these inflammatory responses, it is possible that these changes affected the results with CFCT even in sham-operated mice. The difference in the level of cognitive performance between the two groups would indicate a level of Fg-induced NF-κB p65 involvement in the reduction of overall inflammation caused by m-mTBI. The difference in the cognitive performance between sham-operated mice and the mice with CCI treated with CAPE would suggest that factors other than those induced by an increased interaction of Fg with its receptors, resulting in transcellular transcytosis and extravascular deposition of Fg, may contribute to the level of inflammation during m-mTBI.

Although we found that there was not significantly less DHE/NeuN co-localization in mice with CCI that were administered CAPE compared to the VEH-administered mice with CCI, there was a significant difference in the sham groups where CAPE-treated mice expressed less DHE than the VEH-treated mice. This indicated that the antioxidant capacity of CAPE may be greater in cases of minimal injury, such as in mice with a sham operation, where there was a little damage associated with the creation of the cranial window. Our findings, in agreement with the findings of others [17], demonstrated that CAPE had antioxidant effects on neurons in vitro and in vivo, suggesting its possible remedial usefulness in cases of inflammation caused by m-mTBI or other neuro-inflammatory diseases.

M-mTBI is known to induce the activation of ICAM-1, which could initiate a neuro-inflammatory cascade [40]. During inflammation, pro-inflammatory cytokines such as tumor necrosis factor alpha and interferon gamma are known to induce the activation of neuronal ICAM-1 [41]. We have shown that Fg induced the activation of ICAM-1 in neurons and caused the increased expression of pro-inflammatory cytokines *IL-6*, *CXCL-10*, and *CCL-2* [5]. The presence of extravascular Fg and its interaction with neurons could potentially exacerbate neuro-inflammation indicated by activation of NF-κB p65 [5] and the generation of ROS seen in the present study. Increased activation of neuronal ICAM-1 during m-mTBI enhances the interaction of Fg with neurons and their binding, augmenting the destructive effects. It is known that during the acute period, TBI causes an increase in BBB permeability [18,42,43]. This was shown by the loss of immunoreactivity towards Claudin-5, a marker of endothelial tight junctions, and increased extravasation of injected small molecules, such as Evans Blue, during TBI [18]. The same study demonstrated that treatment with CAPE reduced the TBI-induced BBB permeability and loss of expression of Claudin-5, suggesting that CAPE preserved the endothelial barrier integrity to small molecules [18]. However, the ability of CAPE to protect against increased protein transcytosis was not clear. In general, it is known that large molecules [44] and proteins [45] do not cross BBB. Fg is considered a large protein with a length of about 46 nm [46] and with Stokes radius of 8.4 nm [47]. Present data demonstrate the inability of CAPE to reduce the deposition of Fg during m-mTBI, while CAPE positively reduced the formation of ROS. Thus, these results suggest that while CAPE has the ability to preserve the endothelial junction properties through possible restoration of Claudin-5 expression [18,42,43], it was unable to affect caveolar protein transcytosis, as we have shown previously [3,19,22]. Furthermore, in the present study, we also showed that blocking NF-κB p65 activity with CAPE resulted in a reduction in Fg-associated neurodegeneration. The CAPE reduced the loss of neurons during m-mTBI without reducing the extravascular deposition of Fg (Figure 3 and Appendix A). As we mentioned, our data indicate that CAPE had no significant effect on transcellular transcytosis, the main pathway that is involved in Fg extravasation during inflammation [19]. Yet, we observed reduced co-localization of Fg and DHE in brain tissue of CAPE-treated mice with CCI compared to that in mice with sham operations (Figure 3). This suggests that the protective effect of CAPE was likely due to the inhibition of the NF-κB p65 activity that directly suppresses the generation of ROS [17]. In addition, it has been shown that blocking NF-κB p65 activity directly inhibits neuronal ICAM-1 expression. Thus, it is possible that NF-κB p65 inhibits Fg binding to neurons by reducing neuronal ICAM-1 expression. Collectively, our current and previous data suggest that interfering with a Fg interaction with its receptors directly, or reducing its availability by reducing its blood content, could be advantageous in mitigating its neuro-inflammatory effects [6,13,22].

The limitation of the use of CAPE is its poor water solubility. Among the different studies that evaluated CAPE for its neuroprotective effects, there is a lack of consensus regarding the optimal preparation method for its use as an injectable solution. Future studies should explore the sustained-release delivery of CAPE and characterize the profile of its pharmacokinetic and pharmacodynamic properties to better characterize its effect.

## 5. Conclusions

In conclusion, our data confirm previous findings that CAPE was able to reduce Fg-induced increased generation of ROS and death in neurons, reduced neurodegeneration, and improved cognitive function during m-mTBI. These effects may be due to the specific blocking of Fg-induced NF-κB p65 activity, as we have shown earlier [13].

## Figures and Tables

**Figure 1 biomolecules-15-00080-f001:**
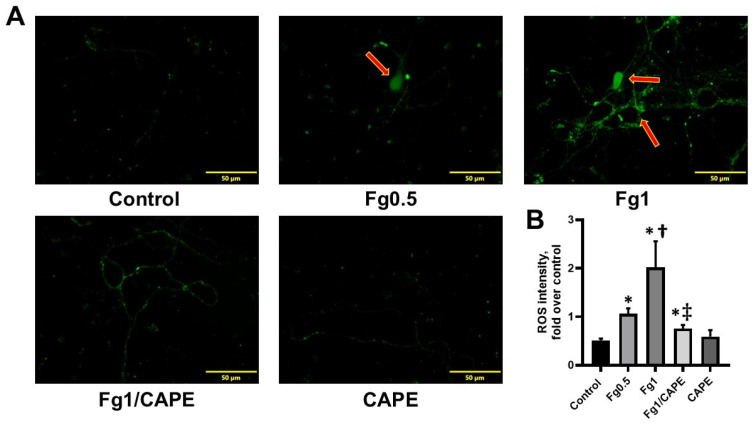
Fibrinogen (Fg)-induced generation of reactive oxygen species (ROS) in cultured neurons. (**A**) Representative images show ROS generation (green) by neurons in response to treatment with phosphate-buffered saline (control), 0.5 mg/mL of Fg (Fg0.5), 1 mg/mL of Fg (Fg1), 1 mg/mL of Fg in the presence of caffeic acid phenethyl ester (CAPE) at 25 µg/mL (Fg1/ CAPE), or with only CAPE. Red arrows point to the ROS generation in the neuron’s soma and axons. (**B**) Summary of analyses of images. *p* < 0.05; *—vs. Control; †—vs. Fg0.5; ‡—vs. Fg1; *n* = 6.

**Figure 2 biomolecules-15-00080-f002:**
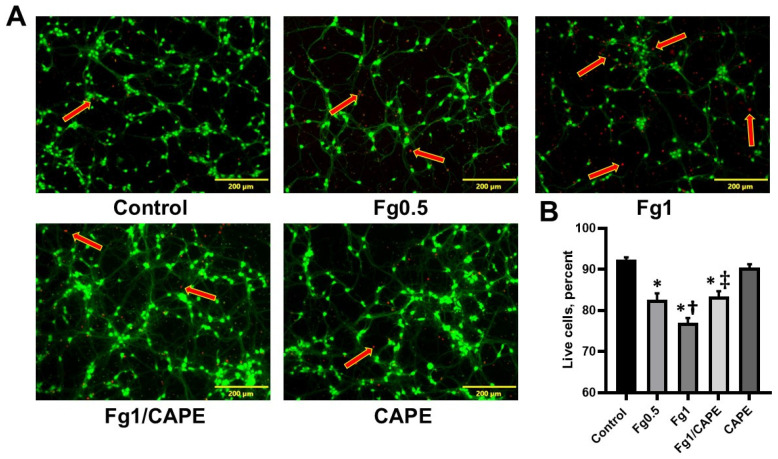
Effect of fibrinogen (Fg) on the viability of cultured neurons. (**A**) Representative images of neurons stained with calcein-AM (green), indicating live cells, and ethidium homodimer (red), indicating dead cells (shown with red arrows). (**B**) Summary of analyses of images. The cells were counted using the automated cell counting function of the CellSens Dimension Desktop software that identified well-defined live cells and quantified them as relative to the total of number of cells in the image. Results were presented as the percentage of the number of live cells of the total cell count. Data were averaged for each group. *p* < 0.05; *—vs. Control; †—vs. Fg0.5; ‡—vs. Fg1; *n* = 6.

**Figure 3 biomolecules-15-00080-f003:**
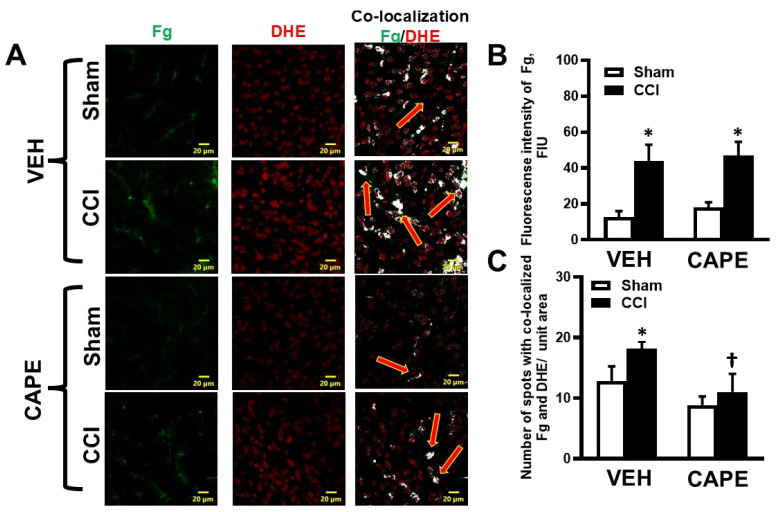
The deposition of fibrinogen (Fg) and the production of reactive oxygen species (ROS) in cortical brain samples assessed by dihydroethidium (DHE) staining 14 days after cortical contusion injury (CCI) in mice treated with caffeic acid phenethyl ester (CAPE) or vehicle (VEH). (**A**) Representative immunofluorescence images with a deposition of Fg (green) and DHE (red), and their co-localization (shown as white dots and indicated with red arrows). Summaries of Fg deposition assessed as fluorescence intensity of Fg and presented as fluorescence intensity units (FIUs) (**B**) and its co-localization with DHE (**C**) were quantified as the number of spots per unit area. *p* < 0.05; *—vs. Sham; †—vs. VEH CCI; *n* = 6.

**Figure 4 biomolecules-15-00080-f004:**
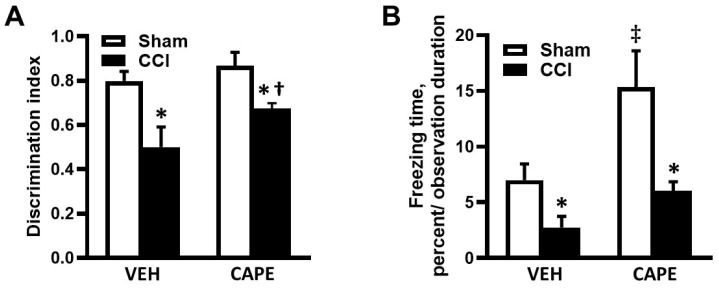
Short-term memory (STM) reduction 14 days after cortical contusion injury (CCI) in mice treated with caffeic acid phenethyl ester (CAPE) or vehicle (VEH). Results of mouse STM were assessed by the (**A**) novel object recognition test and (**B**) contextual fear conditioning test. *p* < 0.05; *—vs. Sham; †—vs. VEH CCI; ‡ vs. VEH Sham; *n* = 10.

## Data Availability

The original contributions presented in this study are included in the article/Appendix A. Further inquiries can be directed to the corresponding author.

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
