# Peer review of "Caffeic Acid Phenethyl Ester Protects Neurons Against Oxidative Stress and Neurodegeneration During Traumatic Brain Injury"

_biomolecules, 2025, doi:10.3390/biom15010080_

Round 1

Reviewer 1 Report

Comments and Suggestions for Authors

The authors aimed to evaluate the effect of caffeic acid phenethyl ester (CAPE), an inhibitor of NF-κB, on Fg-induced neurodegeneration in vitro and in mice with mild-to-moderate TBI.

The manuscript has good scientific facts.

The first keyword may be checked from MeSH database.

In the introduction section, please add the sentence- “CAPE has been reported to have anti-inflammatory, antioxidant, neuroprotective, antiviral, antitumoral, antiatherosclerotic, and immunomodulatory properties in a variety of systems”

Does coagulopathy occur in TBI? Possibly due to a combination of hemorrhage and trauma induced coagulopathy, there can be low fibrinogen levels with severe TBI.

Refer to line 346- Instead of using a general term ‘protect blood-brain integrity’ please use the exact word. Does CAPE effectively reduce blood-brain permeability?

Can CAPE may improve BBB integrity by preserving claudin-5 levels? Please discuss this fact.

What is the effect on hippocampal cell loss?

Regarding the neuroprotection offered by CAPE, can the degree of inflammation influence the results?

Author Response

Please see the attached response to reviewer.

Reviewer 2 Report

Comments and Suggestions for Authors

This manuscript is interesting on the relevant issue of neuroprotection, the study is well designed and the text shows important results; I have only a minor suggestion: The title is too long, the line “The role of fibrinogen” could be deleted as the authors did not demonstrate (P8, L308-318) the role of fibrinogen on their results; also, the term “Caffeic acid” would lead to think that caffeine from coffee is proposed (P14,L455-458), I think that the term should be changed from the title (e.g. “A polyphenol…”)

Comments on the Quality of English Language

No comments.

Author Response

(The authors gave the same response as above.)
